# The Application of Rice Straw with Reduced N Fertilizer Improves the Rice Yield While Decreasing Environmental N Losses in Southern China

Han Liu [1,2,3], Tingting Ma [1,2], Li Wan [1,2], Guopeng Zhou [4], Anfan Zhu [5], Xiaofen Chen [1,2,*] and Jia Liu [1,2,*]

[1] Soil and Fertilizer & Resources and Environment Institute, Jiangxi Academy of Agricultural Sciences, Nanchang 330200, China; liuh5917@163.com (H.L.); x21301102@stu.ahu.edu.cn (T.M.); wanli@jxaas.cn (L.W.)

[2] National Engineering & Technology Research Center for Red Soil Improvement, Key Laboratory of Acidified Soil Amelioration and Utilization, Ministry of Agriculture and Rural Affairs, Nanjing 210008, China

[3] College of Agriculture, Yangtze University, Jingzhou 434025, China

[4] College of Resources and Environment, Anhui Agricultural University, Hefei 230036, China; zhouguopeng@caas.cn

[5] Agricultural Technology Promotion Center of Jiangxi Province, Nanchang 330046, China; zhuanfan999@163.com

* Correspondence: xfchen@jxaas.cn (X.C.); liujia@jxaas.cn (J.L.)

**Abstract:** To investigate the effects of straw residues with reduced nitrogen (N) fertilizer on greenhouse gas (GHG) and N losses in paddy fields, we conducted a field experiment during two growing seasons in paddy rice systems in southern China to evaluate the impacts of the application of straw residues with reduced N fertilizer on rice yield, GHG emissions, and ammonia ($NH_3$) volatilization. The four treatments included N100 (conventional dose of N fertilizer), SN100 (conventional dose of N fertilizer + straw), N60 (60% of the conventional dose of N fertilizer), and SN60 (60% of the conventional dose of N fertilizer + straw). We found that the yield of the SN60 treatment was slightly reduced, but the partial factor productivity of applied N ($PFP_N$) was significantly increased by 63.9% compared to the N100 treatment. At the same N application rate, the application of straw increased soil organic C (SOC), methane ($CH_4$) emissions, carbon dioxide ($CO_2$) emissions, global warming potential (GWP), greenhouse gas intensity (GHGI), and net ecosystem carbon budget (NECB), but significantly decreased soil $N_2O$ emissions and $NH_3$ volatilization. Compared with conventional fertilization (N100), straw residues with reduced N fertilization (SN60) reduced $N_2O$ emissions and $NH_3$ volatilization by 42.1% and 23.9%, and increased GHGI and NECB by 11.1% and 18.3%, respectively. The results indicate that straw residues with reduced N fertilizer are a feasible strategy to reduce N losses in paddy fields while increasing carbon sequestration.

**Keywords:** greenhouse gases; $NH_3$ volatilization; net ecosystem carbon budget; rice yield

## 1. Introduction

China is one of the world's largest rice producers and consumers of nitrogen (N) fertilizer [1,2]. To meet the needs of food for the increasing population, rice yields need to be continuously increased, and the increasing application of N fertilizer is an important way to pursue high rice yields [3]. A meta-analysis showed that the yield could reach a maximum value when the N rate is in the range of 193–250 kg ha$^{-1}$, and there is a risk of yield reduction when exceeding 300 kg ha$^{-1}$ [4]. High inputs of N fertilizer have a positive effect on yields but could lead to a series of environmental pollutions (e.g., decreased soil fertility, increased greenhouse gas (GHG) emissions, and ammonia ($NH_3$) volatilization) [5,6]. Excessive application of N fertilizer would result in soil acidification and reduce soil fertility [7]. In addition, $NH_3$ volatilization is also an important pathway for N losses in paddy soils, with up to 60% of annual N lost to the atmosphere

through $NH_3$ volatilization [8]. Yang et al. [9] found that compared with 100% N fertilizer, reducing N fertilizer by 25% would reduce global warming potential (GWP) and greenhouse gas intensity (GHGI) by 12.1% and 10.3%, respectively. Therefore, exploring reasonable N fertilizer application patterns is of great significance to reduce N losses, improve N fertilizer use efficiency, and reduce GHG emissions and $NH_3$ volatilization.

Rice straw is a common agricultural byproduct with a C/N ratio of 50:1~60:1, and it contains a large amount of organic carbon and nutrients [10]. The application of straw could improve soil fertility, reduce nitrogen loss, and benefit crop yields [3,11]. The application of straw could not only replace part of the chemical N fertilizer but also better solve the problem of straw removal and retention to avoid the waste of resources and gases emitted to the atmosphere by burning [3]. However, in flooded rice fields, the incorporation of straw will provide a large amount of carbon sources for methanogenic microorganisms, which would result in higher methane ($CH_4$) emissions, thereby offsetting the carbon sequestration benefits of straw application [3,5,12]. In addition, many studies have found that the incorporation of straw enhances carbon dioxide ($CO_2$) emissions [13]. Different from the studies on $CH_4$ and $CO_2$, the impact of straw residues on nitrous oxide ($N_2O$) emissions is still unclear. Paddy soil $N_2O$ emissions are mainly produced through nitrification and denitrification [2]. According to a meta-analysis [14], the impact of straw addition on $N_2O$ emissions is primarily influenced by local climate, soil conditions, N fertilizer application amount, the straw C/N ratio, and field management. Many studies have observed that the application of straw would decrease soil oxygen content due to promoting the activities of microorganisms by providing more carbon sources, thereafter creating an anaerobic environment that would increase soil $N_2O$ emissions by denitrification [15,16].

There are different views on the impact of straw residues on $NH_3$ volatilization in rice fields [17]. Several studies have shown that the application of straw and N fertilizer is an obstacle to the fixation of $NH_4^+$-N in the soil and increases the pH of field surface water, which promotes $NH_3$ volatilization [18–20]. On the contrary, some studies revealed that a large amount of humus is formed with the application of straw, which lowers soil pH and enhances the fixation of $NH_4^+$-N in the soil, preventing $NH_3$ volatilization [21].

Previous studies have shown that the application of straw could reduce N fertilizer application by 30% while maintaining crop yields in the next growing season [22]. In addition, the effects of straw residues and reducing N fertilizer application on GWP, GHGI, and net ecosystem carbon budget (NECB) are still unclear. Therefore, we conducted a field experiment during two growing seasons to evaluate the effects of rice yield, GHG emissions, and $NH_3$ volatilization with the application of straw and reduced N fertilizer application in double-cropping paddy rice areas.

## 2. Materials and Methods

### 2.1. Description of the Experimental Site

The field trial took place between April 2023 and October 2023 at the Gao'an experiment base of Jiangxi Academy of Agricultural Sciences in Yichun City, Jiangxi Province (28°15′ N, 115°54′ E). This area experiences a typical subtropical monsoon climate, with an average annual air temperature and precipitation of approximately 17.7 °C and 1600 mm, respectively. The soil in the experimental plot originates from river alluvial deposits and exhibits a red clay texture. The main properties of the soil before the experiment are as follows: pH 5.4, soil organic matter: 19.4 g kg$^{-1}$, total N: 2.0 g kg$^{-1}$, alkaline hydrolyzable-N: 176 mg kg$^{-1}$, available P: 4.3 mg kg$^{-1}$, and available K: 125 mg kg$^{-1}$.

### 2.2. Experimental Design and Field Management

The test was carried out in field test plots in 2023, with an individual plot size of 32 m$^2$ (4 m × 8 m). The plots were separated by soil columns (40 cm × 30 cm) to prevent lateral loss of water and nutrients. Four treatments, including N60 (60% of the conventional dose of N fertilizer); SN60 (60% of the conventional dose of N fertilizer + straw residues); N100 (conventional dose of N fertilizer); and SN100 (conventional dose of N

fertilizer + straw residues), were implemented using a randomized block design, with each treatment replicated three times. After the late rice was harvested in 2021, rice straw was incorporated into the experimental field with an application amount of 6000 kg ha$^{-1}$ [7,23]. The fertilizer management in this experiment was consistent with local farmers. Urea (N: 46%), superphosphate (P$_2$O$_5$: 12%), and potassium chloride (K$_2$O: 60%) were applied as N, P, and K fertilizers. N fertilizer was applied twice, with the ratio of base fertilizer/tiller fertilizer = 6:4; P and K were applied as base fertilizer. The detailed fertilization schedule for early and late rice in 2023 is presented in Table 1.

**Table 1.** The application amount of N, P, K, and rice straw under different treatments in the double-cropping rice system in 2023.

| Treatment | Basal Fertilizer (kg ha$^{-1}$) | | | Tiller Fertilizer (kg ha$^{-1}$) | Rice Straw (kg ha$^{-1}$) |
|---|---|---|---|---|---|
| | N | P | K | N | |
| Early rice | | | | | |
| N60 | 55 | 75 | 120 | 36 | 0 |
| SN60 | 55 | 75 | 120 | 36 | 6000 |
| N100 | 91 | 75 | 120 | 60 | 0 |
| SN100 | 91 | 75 | 120 | 60 | 6000 |
| Late rice | | | | | |
| N60 | 65 | 75 | 150 | 43 | 0 |
| SN60 | 65 | 75 | 150 | 43 | 6000 |
| N100 | 108 | 75 | 150 | 72 | 0 |
| SN100 | 108 | 75 | 150 | 72 | 6000 |

The early rice variety was Zhongjiadao 17, which was manually transplanted on May 1 and harvested on 13 July 2023. Rice was seeded in rows with a spacing of 20 cm between them, and the planting density averaged approximately 20 hills m$^{-2}$. The late rice variety was Yuxiaingyou, which was manually transplanted on 1 August and harvested on 20 October 2023. The sowing method was the same as that of early rice. Field irrigation practices encompass shallow water application during the initial phase of rice growth (tillering stage), drainage and soil drying during the middle phase (heading stage), alternating wet and dry conditions in the later stages, and natural drying subsequent to each irrigation cycle.

## 2.3. Collection and Measurement of Grain and Soil Samples

In both the early and late rice maturity periods, manual harvesting and mechanical threshing were employed in each experimental plot. Subsequently, 200 g of grain from each plot was collected and subjected to drying in an oven at 80 °C. The moisture content of the grain was determined, and the grain yield was adjusted based on the calculated moisture content (14.0%). Prior to rice harvest, soil samples (0~20 cm depth) were obtained from five distinct locations within each plot using a soil auger with a 4 cm diameter. Following the removal of residual roots and debris, the soil samples were air-dried and passed through a 2 mm sieve.

The soil pH was assessed utilizing the potentiometer technique (soil/water ratio of 1:2.5). The soil organic matter (SOM) content was determined using an elemental analyzer (Elementar Analysensysteme GmbH, Hanau, Germany), while the total nitrogen (TN) was quantified via the Kjeldahl nitrogen method [24]. Alkali-hydrolyzable nitrogen (AN) was measured employing an alkali hydrolysis diffusion method [25], available phosphorus (AP) was determined using the sodium bicarbonate extraction–molybdenum antimony colorimetric method, and available potassium (AK) was analyzed via the ammonium acetate extraction–flame photometry method [26].

*2.4. Gas Sampling and Analysis*

GHG fluxes were monitored for the whole rice-growing season using the static chamber approach [27], which contained upper and lower parts. The lower part was a round plastic base frame (with a diameter of 55 cm and a height of 30 cm) and inserted down to a 20 cm soil depth, sowing the same density of rice seedlings in the chamber. Gas samples were collected every 2–3 days after fertilization and irrigation, and every 5–10 days at other times. On the day of sampling, four gas samples were collected with an automatic sampler every 10 min from 9:00 a.m. to 11:30 a.m., and samples were analyzed within 48 h. A gas chromatograph (Agilent 7890B, Agilent Technologies, Santa Clara, CA, USA) was used, equipped with a flame-ionization detector (FID) and electron-capture detector (ECD); $CH_4$ and $CO_2$ were determined by FID at 300 °C; and $N_2O$ was determined by ECD at 350 °C. The gas flux was calculated according to the following formula:

$$F = \rho \times H \times \frac{d_c}{d_t} \times \frac{273}{273 + T} \tag{1}$$

where $F$ ($\mu g \ m^{-2} \ h^{-1}$ for $N_2O$ and $mg \ m^{-2} \ h^{-1}$ for $CH_4$ and $CO_2$) is the gas flux; $\rho$ ($CH_4 = 0.714 \ kg \ m^{-3}$, $CO_2 = 1.977 \ kg \ m^{-3}$, $N_2O = 1.25 \ kg \ m^{-3}$) is the density of GHG under standard conditions; $H$ (m) is the height of the chamber; $\frac{d_c}{d_t}$ ($mL \ m^{-3} \ h^{-1}$) is the rate of change of the GHG concentration per unit time in the chamber; 273 is the temperature conversion coefficient from Celsius to Kelvin; and $T$ (°C) represents the mean temperature within the chamber.

Linear interpolation was employed between observation days to compute daily rates of GHG emissions. The GHG emissions were calculated as follows:

$$f = \sum_{i=1}^{d} \frac{F_i + F_{i+1}}{2} \times (d_{i+1} - d_i) \times 24 \times 10^{-2} \tag{2}$$

where $f$ ($kg \ ha^{-1}$) is the GHG emissions during the rice-growing period, $F_i$ and $F_{i+1}$ are the gas fluxes of the $i$-th and the $i + 1$-th sampling day, $d$ (days) is the difference in days between two adjacent sampling days, and 24 is the number of hours per day.

Over a span of 100 years, the global warming potential (GWP) per unit mass of $CH_4$ and $N_2O$ is 28 and 265 times greater than that of $CO_2$, respectively [28]:

$$GWP\left(kgCO_{2eq}ha^{-1}\right) = N_2O\left(kgN_2Oha^{-1}\right) \times 265 + CH_4\left(kgCH_4ha^{-1}\right) \times 28 + CO_2\left(kgCO_2ha^{-1}\right) \times 1 \tag{3}$$

The greenhouse gas intensity (GHGI) was calculated as follows:

$$GHGI\left(kgCO_2kg^{-1}\right) = GWP/Yield \tag{4}$$

*2.5. Soil NH$_3$ Sampling and Analysis*

$NH_3$ volatilization from the soil was determined by the sponge tracking and KCL extraction methods [29]. The trapping device consisted of a rigid PVC pipe (the diameter was 15 cm and the height was 20 cm) and a ventilated rain cover. Two sponges (2 cm thick and 16 cm in diameter) were spiked with 15 mL of glycerol–phosphoric acid solution (mixed with 50 mL of phosphoric acid, 40 mL of glycerol, and 910 mL of deionized water); the lower sponge (to capture $NH_3$) was placed in the middle of the PVC pipe, and the upper sponge (isolated atmospheric $NH_3$) was flush with the top of the PVC pipe. Samples were collected throughout the whole rice-growing season; samples were taken daily for 10 days after fertilization, and then at intervals of 7–10 days until the rice harvest. The lower sponge of the trapping device was removed at 9:00 a.m. each day and quickly placed in a sealed bag and replaced with two freshly soaked sponges. The lower sponge was placed in a 500 mL plastic bottle with a 1 mol $L^{-1}$ potassium chloride solution of 300 mL and shaken for 1 h. The concentration of ammonium nitrogen in the solution was determined by a

continuous flow analyzer (Skalar San++, Netherlands). The $NH_3$ flux was calculated as follows:

$$v = [M/(A \cdot D)] \cdot 10^{-2} \tag{5}$$

where $v$ (kg ha$^{-1}$ d$^{-1}$) is the $NH_3$ volatilization flux; $M$ (mg) is the amount of ammonia measured by a single device; $A$ (m$^2$) is the cross-sectional area of the trapping device; and $D$ (d) is the number of days for each consecutive trap.

The calculation of $NH_3$ volatilization proceeded as follows:

$$E = \frac{1}{2} \times \sum_{i=1}^{n} [(v_i + v_{i-1}) \times (T_i - T_{i-1})] \tag{6}$$

where $E$ (kg ha$^{-1}$) is the $NH_3$ volatilization; $T_i$ and $T_{i-1}$ represent the duration between two consecutive sampling days.

*2.6. $PFP_N$ and NECB*

The calculation of the partial factor productivity of applied nitrogen ($PFP_N$) proceeded as outlined [30]:

$$PFP_N \left( \text{kg yield kg}^{-1} \text{N} \right) = \frac{Yield \left( \text{kg yield ha}^{-1} \text{season}^{-1} \right)}{N\ fertilizer\ application \left( \text{kg N ha}^{-1}\ \text{season}^{-1} \right)} \tag{7}$$

The net ecosystem carbon budget (NECB) served as a tool to assess the carbon balance within agricultural ecosystems over a short timeframe [30]:

$$NECB = NPP + C_{cover\ crop} - R_h - CH_4 - Harvest \tag{8}$$

$$NPP = NPP_{grain} + NPP_{straw} + NPP_{root} + NPP_{litter} + NPP_{rhizodeposit} \tag{9}$$

where $NPP$ (kg ha$^{-1}$) represents the net primary productivity (NPP) of rice; $C_{straw}$ is the input of rice straw C; $R_h$ (kg ha$^{-1}$) is the carbon emission of soil respiration; $CH_4$ is the methane emission; and $Harvest$ is the carbon content of rice straw and grain.

*2.7. Statistical Analysis*

A two-way analysis of variance (ANOVA) was carried out using the SPSS 26.0 statistical software to analyze the effect of nitrogen fertilizer and straw residues on rice yield, soil physicochemical properties, GHG emissions, GWP, GHGI, NECB, and $NH_3$ volatilization. The figures were prepared using Origin 9.0 (Systat Software Inc., San Jose, CA, USA).

## 3. Results

*3.1. Rice Yield and $PFP_N$*

The yield of early and late rice across various treatments varied from highest to lowest as follows: SN100 > N100 > SN60 > N60. This shows that nitrogen fertilizer could significantly increase the rice yield; however, the addition of straw led to a negligible increase in rice yield (Figure 1a). Compared with the N60 treatment, the yields of N100 and SN60 were higher by 2276 kg ha$^{-1}$ and 391 kg ha$^{-1}$, respectively. The SN100 treatment increased the yield of early and late rice by only 229 kg ha$^{-1}$ compared with the N100 treatment. Unlike the trend observed in rice yield, there was a notable decrease in $PFP_N$ as the nitrogen application rates increased (Figure 1b).

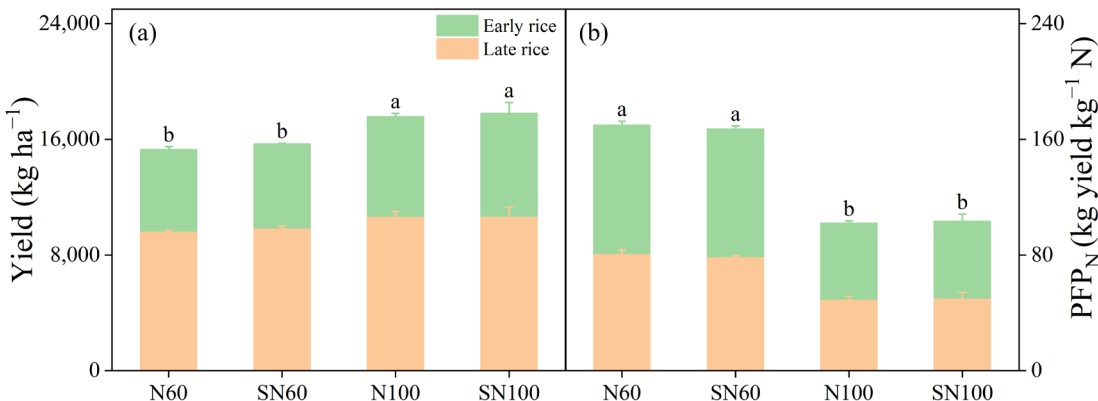

**Figure 1.** Rice yield (**a**) and nitrogen partial factor productivity (PFP$_N$) (**b**) of early rice and late rice for the different treatments investigated. Different lower letters indicate significant differences among treatments ($p < 0.05$). Error bars represent the standard deviation of the mean ($N = 3$).

*3.2. Soil Physicochemical Parameters*

Compared with the N60 and SN60 treatments, the soil pH significantly decreased in the treatments of N100 and SN100 (Table 2). In the late rice season, SOC, AN, and AP in the N100 and SN100 treatments were significantly higher than those in the treatments of N60 and S60. The SOC values in the SN60 treatment were 20.8 g C kg$^{-1}$ and 21.9 g C kg$^{-1}$ in the early rice season and the late rice season, respectively, which were 6.4% and 2.5% higher than those in the N60 treatment. Straw incorporation did not significantly affect soil the TN concentration.

**Table 2.** Soil properties for the different treatments investigated (N = 3).

| Treatment | pH | SOC (g C kg$^{-1}$) | TN (g N kg$^{-1}$) | AN (mg N kg$^{-1}$) | AP (mg P kg$^{-1}$) | AK (mg K kg$^{-1}$) |
|---|---|---|---|---|---|---|
| Early rice | | | | | | |
| N60 | 6.4 ± 0.12 a | 19.5 ± 0.4 b | 2.1 ± 0.20 a | 222 ± 14 a | 11.3 ± 0.2 a | 112 ± 8 b |
| SN60 | 6.4 ± 0.06 a | 20.8 ± 0.2 a | 2.0 ± 0.21 a | 230 ± 15 a | 11.6 ± 0.1 a | 131 ± 5 ab |
| N100 | 6.2 ± 0.09 b | 21.0 ± 0.3 a | 2.1 ± 0.03 a | 241 ± 2 a | 12.3 ± 0.8 a | 115 ± 12 ab |
| SN100 | 6.1 ± 0.03 b | 21.3 ± 0.9 a | 2.2 ± 0.01 a | 256 ± 34 a | 13.2 ± 2.7 a | 133 ± 7 a |
| Late rice | | | | | | |
| N60 | 6.0 ± 0.01 a | 21.4 ± 0.65 b | 2.2 ± 0.08 a | 158 ± 9 b | 8.3 ± 0.1 b | 130 ± 10 a |
| SN60 | 5.9 ± 0.10 a | 21.9 ± 1.0 ab | 2.2 ± 0.16 a | 167 ± 6 b | 9.3 ± 1.0 b | 151 ± 8 a |
| N100 | 5.7 ± 0.05 b | 23.2 ± 0.6 a | 2.3 ± 0.13 a | 205 ± 13 a | 8.7 ± 0.2 b | 131 ± 5 a |
| SN100 | 5.8 ± 0.18 ab | 23.4 ± 0.7 a | 2.4 ± 0.12 a | 207 ± 13 a | 11.5 ± 0.3 a | 151 ± 12 a |

SOC, soil organic carbon; TN, total nitrogen; AN, alkali hydrolyzable nitrogen; AP, available phosphorus; AK, available potassium. Different lower letters indicate significant differences among treatments ($p < 0.05$).

*3.3. CH$_4$, CO$_2$, and N$_2$O Emissions*

The CH$_4$ fluxes showed a similar dynamic for all treatments among the two rice-growing seasons, and two peak fluxes were found in each growing season (Figure 2a). In the early rice season, the highest CH$_4$ fluxes were observed during the middle tillering stage; however, the CH$_4$ fluxes peaked (48.6 mg C m$^{-2}$ h$^{-1}$) after rice transplanting in the late rice season. The treatments with straw (SN60, SN100) had higher CH$_4$ fluxes than those without straw addition (N60, N100) in the early and middle stages of the rice-growing season; the CH$_4$ fluxes were close to zero in all treatments at maturity. The annual CH$_4$ emissions in SN60 and SN100 were 412 and 465 kg C ha$^{-1}$, respectively, which were significantly higher than those in the N60 (272 kg C ha$^{-1}$) and N100 (335 kg C ha$^{-1}$) treatments (Table 2). Moreover, similar results were found for CH$_4$ emissions in all treatments in both the early and late rice seasons.

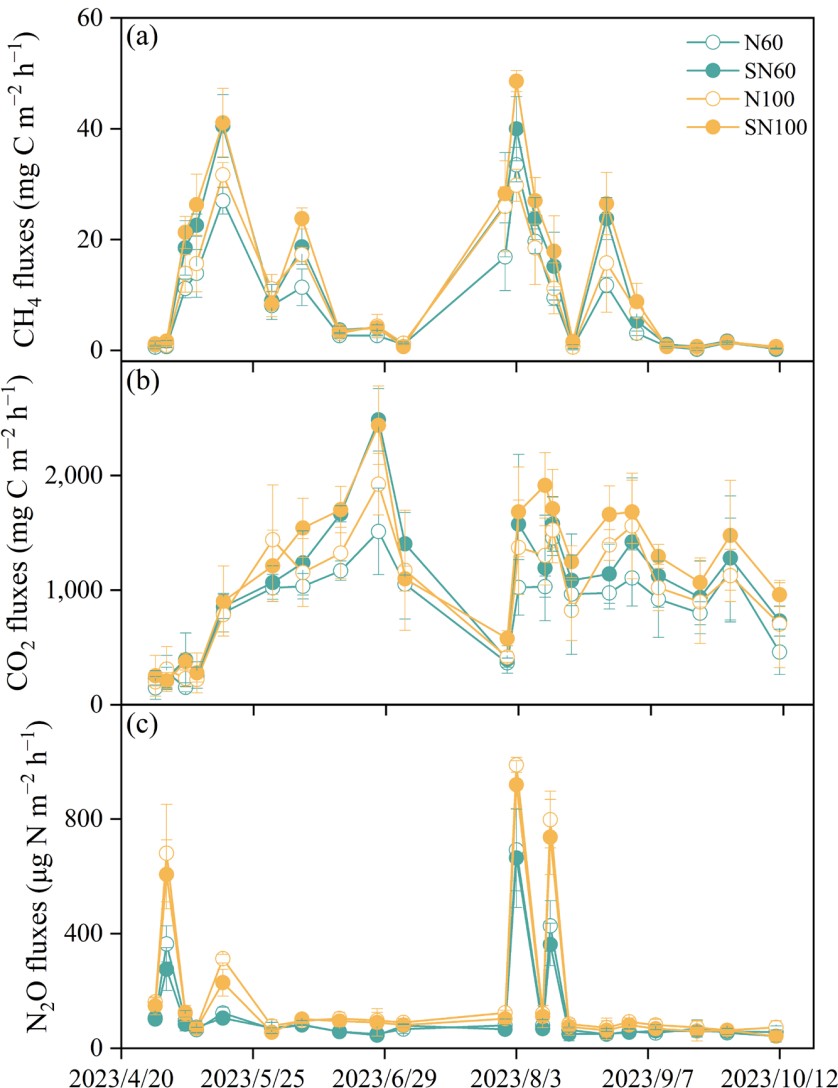

**Figure 2.** Daily mean $CH_4$, $CO_2$, and $N_2O$ fluxes (**a**–**c**) during the early rice and late rice seasons. Error bars represent the standard deviation of the mean ($N = 3$).

The $CO_2$ fluxes showed an increasing tendency in the early rice season while showing a fluctuating up-and-down tendency in the late rice season (Figure 2b). The $CO_2$ fluxes gradually increased ten days after transplanting the early rice and reached the peak flux 58 days after transplanting (spike stage). In the late rice season, the $CO_2$ fluxes reached a maximum peak 10 days after transplanting, and two subpeaks occurred 32 and 58 days after transplanting. The annual $CO_2$ emissions in SN60 and SN100 were 38,449 kg C ha$^{-1}$ and 43,922 kg C ha$^{-1}$, respectively, which were significantly higher than in N60 (30,744 kg C ha$^{-1}$) and N100 (36,218 kg C ha$^{-1}$) (Table 2). With the application of the same amount of straw residues in the rice paddy field, the $CO_2$ emissions increased with the increasing application of nitrogen fertilizer.

The peak $N_2O$ fluxes were observed to coincide with the fertilization, and the $N_2O$ fluxes were higher in the late rice season than in the early rice season (Figure 2c). The peak $N_2O$ fluxes of each treatment were found after basal fertilization and tillering fertilization, and $N_2O$ fluxes significantly increased with increasing N fertilizer inputs and aggravated gaseous N emissions. The annual $N_2O$ emissions in the N100 and SN100 treatments were 5.7 kg N ha$^{-1}$ and 5.0 kg N ha$^{-1}$, respectively, which were significantly higher than in the N60 (3.6 kg ha$^{-1}$) and SN60 (3.3 kg ha$^{-1}$) treatments (Table 2). In addition, at the same N fertilizer application rate, the incorporation of straw increased the $N_2O$ emissions by 6.4% and 12.0% for SN60 and SN100, respectively.

### 3.4. GWP, GHGI, and NECB

Both straw residue and N fertilizer application had significant effects on GWP. Compared with the SN100 treatment, the N100 and SN60 treatments significantly reduced the GWP by 19.1% and 12.7%, respectively (Table 3). There was no significant difference in GWP between the SN60 and N100 treatments. Similarly, the application of straw and N fertilizer had significant effects on GHGI. As compared to the SN100 treatment, the N100 treatment significantly reduced GHGI by 15.2%, and the SN60 treatment significantly reduced GWP by 12.7%. Straw residues had a significant effect on NECB. Straw addition (SN60, SN100) significantly increased the NECB by 16.9–18.3% compared with no straw addition (N60, N100). Furthermore, the NECB in the SN60 treatment was 3648 kg ha$^{-1}$, which was higher than in the SN100 treatment (33,120 kg ha$^{-1}$).

**Table 3.** Mean cumulative $CH_4$, $CO_2$, and $N_2O$ emissions, GHGI, and NECB among different treatments (N = 3).

| Treatment | $CH_4$ (kg C ha$^{-1}$) | $CO_2$ (kg C ha$^{-1}$) | $N_2O$ (kg N ha$^{-1}$) | GWP (kg $CO_2$-eq ha$^{-1}$) | GHGI (kg $CO_2$-eq kg$^{-1}$) | NECB (kg ha$^{-1}$) |
|---|---|---|---|---|---|---|
| N60 | 272 ± 16 d | 30,744 ± 1331 c | 3.6 ± 0.1 c | 39,307 ± 1607 c | 2.4 ± 0.1 c | 3085 ± 369 c |
| SN60 | 412 ± 13 b | 38,449 ± 2696 b | 3.3 ± 0.2 c | 50,868 ± 2912 b | 3.0 ± 0.2 a | 3648 ± 904 a |
| N100 | 335 ± 38 c | 36,218 ± 1130 b | 5.7 ± 0.3 a | 47,127 ± 2263 b | 2.7 ± 0.1 b | 2840 ± 585 b |
| SN100 | 465 ± 9 a | 43,922 ± 1539 a | 5.0 ± 0.2 b | 58,273 ± 1734 a | 3.2 ± 0.1 a | 3320 ± 104 a |

$CH_4$, methane; $CO_2$, carbon dioxide; $N_2O$, nitrous oxide; GWP, global warming potential; GHGI, greenhouse gas intensity; NECB, net ecosystem carbon budget. Different lower letters indicate significant differences among treatments ($p < 0.05$).

### 3.5. Soil $NH_3$ Volatilization

The $NH_3$ volatilization fluxes in all treatments had a similar dynamic, and $NH_3$ fluxes significantly increased after the application of base fertilizer and tillering fertilizer, and the peak $NH_3$ fluxes in the late rice season were significantly higher than those in the early rice season (Figure 3a). The $NH_3$ fluxes of each treatment during the base fertilizer and tillering fertilizer periods were as follows: N100 > SN100 > N60 > SN60. Both the addition of straw and the application of N fertilizer exerted notable impacts on $NH_3$ volatilization emissions in the double-cropping rice system (Figure 3b). The lowest $NH_3$ volatilization was found in the SN60 treatment, which was 12.6%, 23.9%, and 24.5% lower compared with the N60, N100, and SN100 treatments, respectively.

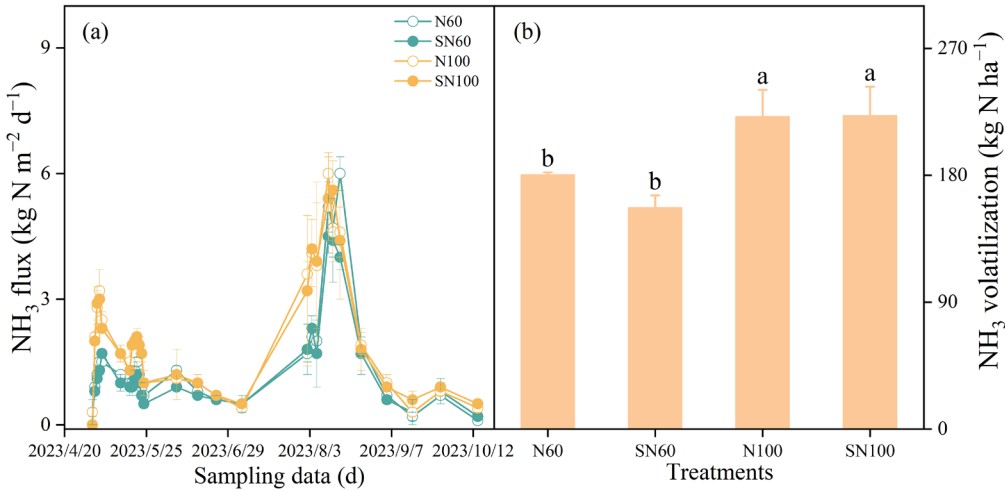

**Figure 3.** Dynamics of soil $NH_3$ fluxes (**a**) and $NH_3$ volatilization (**b**) under different treatments. Different lower letters indicate significant differences among treatments ($p < 0.05$). Error bars represent the standard deviation of the mean ($N = 3$).

## 4. Discussion

### 4.1. Effect of Straw Residues with Reduced N Fertilizer on Soil Fertility and Rice Yield

Straw residues are currently a common field management, which not only improves soil fertility but also ensures the sustainability of agricultural production [31,32]. In our study, the SOC slightly increased with the straw application; this is consistent with many studies that have shown that [7,33–35] the application of straw increased soil SOC, AN, AP, and AK concentrations. The straw residues provide a large carbon source for microorganisms, and the application of nitrogen fertilizer accelerates the decomposition and utilization of straw by the microorganisms [33]. Generally, the SOC is a key factor in evaluating soil fertility [36] and affects crop yield [37].

According to Li et al. [30], incorporating straw can lead to a reduction in N fertilizer usage by 10~30%, while still maintaining rice yield and enhancing the partial factor productivity of nitrogen fertilizer. In contrast to conventional N fertilizer application (N100), there was no notable variance in yield observed with straw residues combined with a 40% reduction in N fertilizer (SN60), while the partial factor productivity of nitrogen fertilizer experienced a significant increase of 54.7% (Figure 1b), as indicated by the findings of this study. The reasons are as follows: (1) This experiment is a long-term experiment carried out from 2015; the long-term application of straw improved the soil fertility and the soil nutrients stayed at a high value [38,39]. (2) Rice straw has a high C/N ratio, and adding nitrogen fertilizer can reduce the C/N ratio in the soil; a reasonable C/N ratio can enhance soil microorganisms and soil enzyme activities, thereby accelerating the decomposition of straw and releasing nutrients into the soil. This explanation is consistent with Liu et al. [40].

### 4.2. Effect of Straw Residues with Reduced N Fertilizer on GHG, GWP, GHGI, and NECB

Both straw residue and N fertilizer application significantly influenced $CH_4$ and $CO_2$ emissions [14,40]. In this study, the $CH_4$ and $CO_2$ emissions increased greatly when increasing the N fertilizer application rate. Increased N application also enhances rice growth and the formation of well-developed aerenchyma, thereby increasing $CH_4$ emissions [41,42]. Moreover, the application of straw also stimulated $CH_4$ emissions, which was consistent with the results of Wang [3]. Straw residues provide abundant metabolic carbon substrates (cellulose, hemicellulose, etc.) [43] for methanogenic bacteria, which is favorable for $CH_4$ emissions. During the early stages of straw decomposition, the low oxygen content in the soil favors methanogenic bacteria, leading to an increase in $CH_4$ emissions [44,45]. Muhammad et al. [46] and Wang et al. [47] found that the application of straw increased soil MBC and MBN, enhanced soil microbial activity, and tended to exacerbate $CO_2$ emissions. In particular, $CO_2$ emissions caused by straw returning are affected by many factors such as soil moisture, soil temperature, and the soil C/N ratio [48].

The nitrification and denitrification processes of soil microorganisms are the main mechanisms for $N_2O$ production in rice fields [48]. The outcomes of our study revealed that the treatments with straw application significantly reduced $N_2O$ emissions compared to the treatments without straw. This aligns with the findings of Shi et al. [49]: that the application of straw has the potential to reduce $N_2O$ emissions due to reducing the application rate of nitrogen fertilization. There were two reasons explaining why straw residues suppress $N_2O$ emissions. Firstly, rice straw has a high C/N ratio, and microorganisms absorb available nitrogen from the soil and reduce the reaction substrate of $N_2O$ during the process of organic matter mineralization due to nitrogen deficiency in the soil [50]. Secondly, the application of straw fills the soil pores and decreases the oxygen content in the soil, which promotes denitrification, and $N_2O$ is reduced to $N_2$ [51].

There is a trade-off between $CH_4$ and $N_2O$ in rice fields [7]. Therefore, to clarify the effects of straw residues with decreased N fertilizer on greenhouse gases, it is necessary to comprehensively consider GWP, GHGI, and NECB. The SN60 treatment increased the GWP by 7.94% and the GHGI by 11.11% compared to the N100 treatment. Considering that we added $CO_2$ emissions when calculating the GWP and GHGI, and that straw residues would inevitably increase $CO_2$ emissions, there is a slight increase in GWP and GHGI.

Positive NECB values indicate that the soil is a carbon sink (i.e., absorbs carbon) and negative NECB values indicate that the soil is a carbon source (i.e., loses carbon) [52,53]. We found that the NECB was positive for all treatments, and that the paddy ecosystem was mainly a carbon sink. The NECB value of the SN60 treatment was greater than that of other treatments; there was a higher carbon input with the addition of straw. Therefore, a reduction in nitrogen fertilizer within a certain range will increase the crop uptake.

*4.3. Effect of Straw Residues with Reduced N Fertilizer on $NH_3$ Volatilization*

$NH_3$ volatilization stands as a primary avenue for nitrogen loss from paddy soils, impacting the productivity of paddy rice systems and nitrogen utilization efficiency [54]. Nitrogen application, floodwater pH, and floodwater $NH_4^+$-N levels constitute the primary factors influencing $NH_3$ volatilization from paddy fields, and decreasing the amount of nitrogen fertilizer applied has been shown to mitigate $NH_3$ volatilization from these fields [55]. Our study demonstrated a significant decrease in $NH_3$ volatilization with the reduction in nitrogen fertilizer (Figure 3), consistent with findings from prior research [54]. $NH_3$ volatilization was 41.0% lower in the straw residues with reduced N fertilizer treatment (SN60) than those with conventional N fertilizer treatment (N100) because we covered the straw directly on the ground surface, which had a weak stimulating effect on the soil urease activity and slowed down $NH_3$ volatilization. In addition, the mineralization of organic nitrogen in straw needs more time, forming humus and increasing soil the adsorption capacity, which inhibits $NH_3$ volatilization. Xia et al. [56] discovered that incorporating urease inhibitors and straw led to a notable decrease in the $NH_4^+$-N concentration in floodwater, consequently mitigating $NH_3$ volatilization. Similarly, Xu et al. [5] observed that utilizing domestic sewage water for irrigation resulted in reduced $NH_4^+$-N concentration and pH levels in floodwater, thereby slowing down $NH_3$ volatilization. Overall, the application of rice straw with reduced N fertilizer provides an important reference for us to improve the utilization rate of N fertilizer and reduce soil $NH_3$ volatilization.

**5. Conclusions**

By comparing the treatment of rice straw and/or reduced N fertilizer in double-cropping rice systems, we examined the rice yield, soil physicochemical parameters, GHG emissions, and soil $NH_3$ volatilization for two growing seasons. The SN60 treatment could maintain the yield while significantly increasing the $PFP_N$. The application of straw increased the soil available nutrients and soil fertility, with the SOC being significantly higher in treatments with straw application than those in treatments without straw application. The SN60 treatment increased $CH_4$ and $CO_2$ emissions due to the addition of large C sources, while the reduction in N fertilizer significantly decreased $N_2O$ emissions and soil $NH_3$ volatilization and can be regarded as a recommended treatment.

**Author Contributions:** X.C. and J.L. designed the experiments. H.L., T.M., A.Z. and X.C. completed the field sampling. H.L. and L.W. performed the data analysis and prepared the figures. H.L. wrote the manuscript. H.L., L.W., G.Z. and J.L. contributed to the revision of the manuscript. All authors have read and agreed to the published version of the manuscript.

**Funding:** This research was supported by the Special Program for Basic Research and Talent Training of the Jiangxi Academy of Agricultural Sciences (JXSNKYJCRC202425); the Innovation Fund of the Jiangxi Academy of Agricultural Sciences, China (20182CBS002); and the National Natural Science Foundation of China (42267046, 32160766).

**Institutional Review Board Statement:** Not applicable.

**Informed Consent Statement:** Not applicable.

**Data Availability Statement:** Data are available from the authors.

**Conflicts of Interest:** The authors declare no conflict of interest.

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
