# Peer review of "The Application of Rice Straw with Reduced N Fertilizer Improves the Rice Yield While Decreasing Environmental N Losses in Southern China"

_sustainability, doi:10.3390/su16072737_

Round 1
Reviewer 1 Report
Comments and Suggestions for Authors
Comments on the Quality of English LanguageMinor editing of English language required
Author Response
Dear Editor and anonymous reviewers,
Thank you very much for reviewing our manuscript submitted to Sustainability (Manuscript ID: 2930846) and providing us with many constructive comments, which have significantly improved the quality of our work. The manuscript has been revised from the previous submission, following comments and suggestions from the editor and the three reviewers. Below, you will find a detailed accounting of how each comment was addressed.
Thank you for your time and consideration.
Yours sincerely,
Han Liu and Xiaofen Chen
March 22, 2024
----------------------------------------------------
Point-by-point responses to the reviewers.
Response to Reviewer #1:
- Line 16 correct the format.
Revised accordingly
- Format isn’t uniform throughout the text
We are sorry for our carelessness. We checked and corrected throughout the manuscript.
- Results (not shown) can be removed from the text.
Deleted.
- Straw returning term must be replaced with some appropriate term such as straw
residues etc,if possible
We used “the application of straw” or “straw residues” to replace “straw returning”.
- Hypothesis and need or objectives for the study must be clarified.
Thanks for your constructive comments. We revised the introduction part.
- Relationship of different parameters can better help to understand the results either by correlation or using 2way ANOVA giving overall impact of treatments
We agreed with the reviewer’s comment. We did the relationship analysis and two-way ANOVA, the results showed the same as we put the difference in the figures and tables.
- Line 402-415 discussion must be revised.
We rewrote the paragraph in the revised manuscript. See lines 413-430.
- Must be revised giving future direction and gaps of the study. Line 499-500 must be eliminated.
We reorganized the part and recommended the application of straw residues with reduced N fertilizer in the double-cropping rice field.

Reviewer 2 Report
Comments and Suggestions for Authors
The research subject is interesting and timely. The publication written exceptionally carefully. Few remarks are marked in the text. These are mainly comments on editing. The methodology is well described. The notation of abbreviations of journal titles should be corrected in references. Each word given as an abbreviation should be followed by a dot.

Author Response
Dear Editor and anonymous reviewers,
Thank you very much for reviewing our manuscript submitted to Sustainability (Manuscript ID: 2930846) and providing us with many constructive comments, which have significantly improved the quality of our work. The manuscript has been revised from the previous submission, following comments and suggestions from the editor and the three reviewers. Below, you will find a detailed accounting of how each comment was addressed.
Thank you for your time and consideration.
Yours sincerely,
Han Liu and Xiaofen Chen
March 22, 2024
----------------------------------------------------
Response to Reviewer #2
- No e-mail addresses for the other authors.
Added accordingly.
- Abstract is too long.
Thanks for this helpful suggestion. The abstract has been shortened. See lines 20-42 (hereafter, line number was referred to the clean version manuscript).
- This position no2 should be removed.
Revised accordingly.
- Greenhouse gases have been written about 2 times. Correction.
Thanks for pointing out this error, we have deleted the sentence “Similarly, large amounts of N fertilizer application would increase soil GHG emissions, especially N2O.”.
- Give the entire name of the abbreviation on the first entry.
We have given the entire name of the abbreviation on the first entry throughout the manuscript.
- space Correct anywhere spaces are missing.
Revised accordingly throughout the manuscript.
- Does this article describe the method of analysis used?
Yes, we added the relevant references for the analysis method.
- Correct the notation of item 24 in references.
Revised accordingly.
- Place this item in references.
Revised accordingly.
10. Provide short information on Origin 9.0
Added accordingly.
- large font
Revised accordingly.
- Is the unit here definitely grams? Add an explanation of abbreviations under the table.
Yes, it’s grams. We carefully checked the units of SOC and TN which both are g kg-1, and added explanations of abbreviations under the table.
- Add an explanation of abbreviations under the table.
Added accordingly.
- State who concluded this. In the experiment or another author.
Added accordingly.
- This is just a summary. This position should be removed
Revised accordingly.
- What the dates are. Correction.
Revised accordingly.

Reviewer 3 Report
Comments and Suggestions for Authors
The scientific manuscript shows scientific quality and merit for publication in Sustainability.
Corrections and/or suggestions are in the attached file.

Author Response
Dear Editor and anonymous reviewers,
Thank you very much for reviewing our manuscript submitted to Sustainability (Manuscript ID: 2930846) and providing us with many constructive comments, which have significantly improved the quality of our work. The manuscript has been revised from the previous submission, following comments and suggestions from the editor and the three reviewers. Below, you will find a detailed accounting of how each comment was addressed.
Thank you for your time and consideration.
Yours sincerely,
Han Liu and Xiaofen Chen
March 22, 2024
----------------------------------------------------
Response to Reviewer #3
- Insert the Aims of study.
Thanks for this helpful suggestion. We have added the aims of the study to the abstract. See lines 20-22.
- Delete word shown in title
Deleted.
- Indicate the dose of N applied, commonly for high productions.
We added the number of doses of N fertilizer for high production. See lines 52-54.
- Indicate, on average, the composition of nutrients and organic carbon.
Revised accordingly. See lines 70-72.
- Be clearer and more direct in describing the AIMS of the study.
Revised accordingly. See lines 110-114.
- Why did this amount of straw return to the field? Indicate reference of the methodology or justification.
We applied the same amount of straw according to the other studies (Liu et al., 2019; Zhang et al., 2021) in the same area.
- Insert the legend with the statistical p.
Revised accordingly.
- This must be indicated in the figure/table subtitle.
Added accordingly.
- Make it very clear whether these values are serious, in terms of the environment, or are within acceptable limits.
Revised accordingly.
- Make it very clear whether these values are serious, in terms of the environment, or are within acceptable limits.
Added accordingly. See lines 346-347.
- Delete the text, if you don't have the information, it's not interesting to comment.
Deleted.
- Answer that question. Even though gas emissions occurred, it was harmful to the environment
The application of straw residues would result in GHG emissions, but straw return could reduce the application rate of fertilizers which better improves soil, hydrology, and atmospheric health. In our study, as compared with conventional practice (N100), straw return with reduced N fertilizer (SN60) treatment significantly reduced N2O emissions which benefits for environment.
- Be direct and describe what was the best recommended treatment.
Revised accordingly.
